# A Novel Interval Energy-Forecasting Method for Sustainable Building Management Based on Deep Learning

**Yun Duan**

School of Law, Huazhong University of Science and Technology, Wuhan 430074, China; dyun0426@163.com

**Abstract:** Energy conservation in buildings has increasingly become a hot issue for the Chinese government. Compared to deterministic load prediction, probabilistic load forecasting is more suitable for long-term planning and management of building energy consumption. In this study, we propose a probabilistic load-forecasting method for daily and weekly indoor load. The methodology is based on the long short-term memory (LSTM) model and penalized quantile regression (PQR). A comprehensive analysis for a time period of a year is conducted using the proposed method, and back propagation neural networks (BPNN), support vector machine (SVM), and random forest are applied as reference models. Point prediction as well as interval prediction are adopted to roundly test the prediction performance of the proposed model. Results show that LSTM-PQR has superior performance over the other three models and has improvements ranging from 6.4% to 20.9% for PICP compared with other models. This work indicates that the proposed method fits well with probabilistic load forecasting, which could promise to guide the management of building sustainability in a future carbon neutral scenario.

**Keywords:** probabilistic load forecasting; LSTM; penalized quantile regression; interval prediction

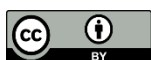

## 1. Introduction

Energy consumption in the Chinese office building sector has reached nearly one-fifth of China's total energy consumption in 2015 [1]. This makes the study of energy use of buildings an essential part of the implementation of energy conservation policies in China. Moreover, with the development of information technology, extensive building operation data are available to be recorded and collected. It, therefore, allows the data-driven methods to be widely applied in the field of building energy efficiency enhancement. Among the different data driven applications, building energy consumption prediction is the most widespread one [2].

In general, the time horizons of office building energy prediction can be roughly classified into three categories, that is, short term for one day or less, medium term for one week to one month and long term for one year. Due to the applications in real-time control and anomaly detection, short-term forecasting has aroused great interest among researchers, while the study of medium- and long-term forecasting has received far less attention. Since the aim of the medium- and long-term forecasting study is to assess building energy consumption and formulate energy plans, ensuring the accuracy of the forecasting results is a priority at the current stage, which is also a fairly difficult objective [3].

Most interval forecasting is presented in the form of probabilistic density function, and one of the most commonly used methods in this field is quantile regression (QR) [4]. Quantile regression has been mainly used in the field of risk assessment and electricity price forecasting, and has shown good results in literature [5–7]. However, to the best of the author's knowledge, few papers are available in the technical literature considering the possibility of application in the field of building energy consumption prediction. In

Ref. [8], D-vine copula quantile regression is applied to predict conditional quantile of heating energy consumption after retrofitting, and it is the first to use quantile regression to analyze residential building energy consumption.

The main contributions of this paper can be summarized as the following.: (1) analyze the prediction performance for daily and weekly indoor load in an office building by combining the deep-learning method with penalized quantile regression, as quantile regression has so far not been adequately studied in the field of building energy consumption prediction; (2) verify the advantages of deep learning in medium- and long-term forecasting by choosing BPNN, SVM, and random forest as the basic point estimation models.

The remainder of this paper is organized as follows. Section 2 reviews the building energy prediction methods. Section 3 introduces the data set and the research synopsis of this article and the LSTM model as well as the penalized quantile regression method used in the study. The results and discussion are detailed in Section 4. Concluding remarks are presented at the end of the paper.

## 2. Literature Review

### 2.1. Related Work for Point Forecasting

The development of short-term forecasting based on the data-driven method evolved from simplicity to complexity. In the past decades, the commonly used machine-learning method, i.e., artificial neural network (ANN) [9], SVM [10], decision tree (DT) [11] have been extensively studied in the literature. For example, Gonzalez and Zamarreno [12] predicted hourly energy consumption with a feedback neural network, and it improved the prediction performance of the traditional ANN. Chen et.al [13] proposed a short-term load prediction method based on the support vector regression (SVR) model and compared it with the other seven traditional prediction models; results showed that the prediction accuracy of the SVR model is much better than the other models. With the deepening of theoretical research, optimization algorithms have been applied to optimize the parameter selection of the data-driven methods. Common algorithms include particle swarm optimization (PSO) [14], principal component analysis (PCA) [15], differential evolution (DE) [16], and fuzzy methods [17], and most research so far has corroborated the superiority of optimization algorithms in improving predictive performance.

More recently, deep-learning methods are also introduced in the field of short-term forecasting [18]. By their ascendancy in two aspects, predictive practicability and performance and deep learning promptly became research hotspots. Fu [19] used a deep-belief-network-based method to predict cooling load in real buildings hours ahead, and the results indicate that the deep-belief network performed the best when compared to the prediction algorithms, such as persistence, BPNN, and SVM. Li and Liu [20] adopted deep recurrent neural networks to make the real-time state estimation of room-cooling load. In [21], the deep-learning method is compared with eight commonly used machine-learning methods, authors make a comprehensive analysis of the day-ahead energy demand prediction, and infer that the deep-learning method outperforms other popular traditional machine-learning methods with higher accuracy.

On the whole, the study of short-term forecasting is relatively mature, and the existing research findings can reach the level of engineering application. Despite novel algorithms popping up all the time, the core issue lies in improving accuracy and essentially reducing computational costs. Table 1 exhibits the time horizon and methods used in short-term forecasting articles in recent years, and it can be concluded that most of the short-term forecasting focuses on the day-ahead prediction problem at different time intervals. As pointed out above, short-term forecasting can be regarded as the key link of building energy savings, and with the rapid development of artificial intelligence, it will receive more and more attention.

**Table 1.** The statistics of the short-term forecasting articles.

| Reference | Temporal Granularity | Time Horizon | Method |
|---|---|---|---|
| [22–24] | Hourly | 24 h | ARX, SRWNN,LSTM |
| [25] | 15 min | 24 h | ANN |
| [26] | 5 min | 24 h | mbCRT |
| [27] | 15 min | 15–60 min | Markov model, ANN, SVR |
| [28] | Hourly | 6 h | AR, WT-AR, NM-AR |

Unlike short-term forecasting, of which the prediction accuracy can reach more than 90%, previous literature shows that the relative errors corresponding to medium- and long-term forecasting at hourly resolutions often are in excess of 40% [29–32]. Several studies have focused on forecasting the aggregate building energy usage to improve prediction performance. Williams and Gomez [33] found that monthly energy consumption at the household level can be predicted with 74% accuracy, while for aggregate monthly energy consumption, the prediction accuracy can reach nearly 94%. Cai et al. [34] divided the monthly electricity energy consumption of residential buildings into different categories, and the prediction accuracy of the proposed model vastly exceeds those of conventional methods.

*2.2. Related Work for Interval Forecasting*

The above literature belongs to the application scope of the point estimation methods, which are suitable for the application of real-time control based on short-term forecasting. However, it is impossible to find a high-accuracy and reliability point estimation method in medium- and long-term forecasting, as there exists too many uncertain factors in the medium- and long-term operations of the systems to affect their energy consumption. Therefore, interval forecasting has become increasingly important in recent years, as interval forecasting provides prediction range and confidence rather than a point estimation, which would be more meaningful to the building managers compared with the point estimation. Walter et al. [35] carried out the interval prediction of energy consumption for 17 different commercial buildings and transformed the determined prediction information into the uncertain interval estimation, the authors believe that this interval estimation can provide reliable decision information for the investment of energy-saving measures. Ahmad and Chen [36] studied the problem of the smart grid energy demand prediction, and they predicted the energy demand one month, one quarter, and one year ahead, respectively. They gave a high-precision prediction range and provided a reference for managers to optimize resource planning. Martinez, Soto, and Jentsch [37] proposed a residential energy model that can be applied to different countries. The model can predict the monthly and annual residential energy demand of a given country or region and give the approximate energy consumption in different confidence regions.

**3. Methodology**

*3.1. Data Description*

The data used in this study comprises the temperature, humidity, and flow rate of air inlet and outlet of all air-conditioning units in an office building from Shanghai, China, and the indoor load can be calculated by these data. In addition, we used the *worldmet* package in the open-source software R to obtain the meteorological data from the airport which is near the target building, and the distance between the target building and the corresponding airport is no more than 7 km. Therefore, the meteorological data of the airport can be used as the actual meteorological data of the target building. All the data set were recorded with 15 min resolution from 17 April 2017 to 3 May 2018. In this paper, we considered that LSTM can learn the order dependence between items in a sequence,

and it has the promise of being able to learn the context required to make predictions in time-series forecasting problems. Therefore, we ignore the date information of the input variables in the expectation of further demonstrating the predictive advantage of the LSTM model.

In total, the dataset contains four different variables, each with 36,672 observations. The raw data set is divided into two parts, training set (70%) and testing set (30%), the former consists of the indoor load and weather data from April 2017 to January 2018, and the latter contains the rest of the data set. Meanwhile, missing values are processed by interpolation, and outliers do not need to be removed or replaced, as there is no explicit outlier label in the data set, and some outlier detection methods based on statistical rules may destroy the authenticity of the data set.

Figure 1 presents the probability distribution of the indoor load and weather data. The probability distribution of the indoor load exhibited in Figure 1A reveals that it is close to skewed normally distributed. In addition, it can be concluded from other subgraphs that the weather in Shanghai may be humid and hot in summer, which is in line with its subtropical monsoon climate characteristics. The occasional extreme weather in Figure 1B could cause a sharp increase in indoor load, and this situation is also reflected in Figure 1A. In addition, the most likely distributions of each dataset in Figure 1 are evaluated by the commonly used Kolmogorov–Smirnov test, and the results are shown in Table 2:

**Table 2.** Distribution analysis of the indoor load and weather data.

| Data | Distribution | Parameters |
|---|---|---|
| Energy consumption | Lognormal | Mean of logarithmic values: 4.284 |
| | | Standard deviation of logarithmic values: 0.706 |
| Ambient temperature | Normal | Mean:18.481 |
| | | Standard deviation: 9.451 |
| Relative humidity | Extreme value | Location parameter: 77.860 |
| | | Scale parameter:14.660 |
| Wind speed | Weibull | Scale parameter: 4.564 |
| | | Shape parameter: 2.473 |

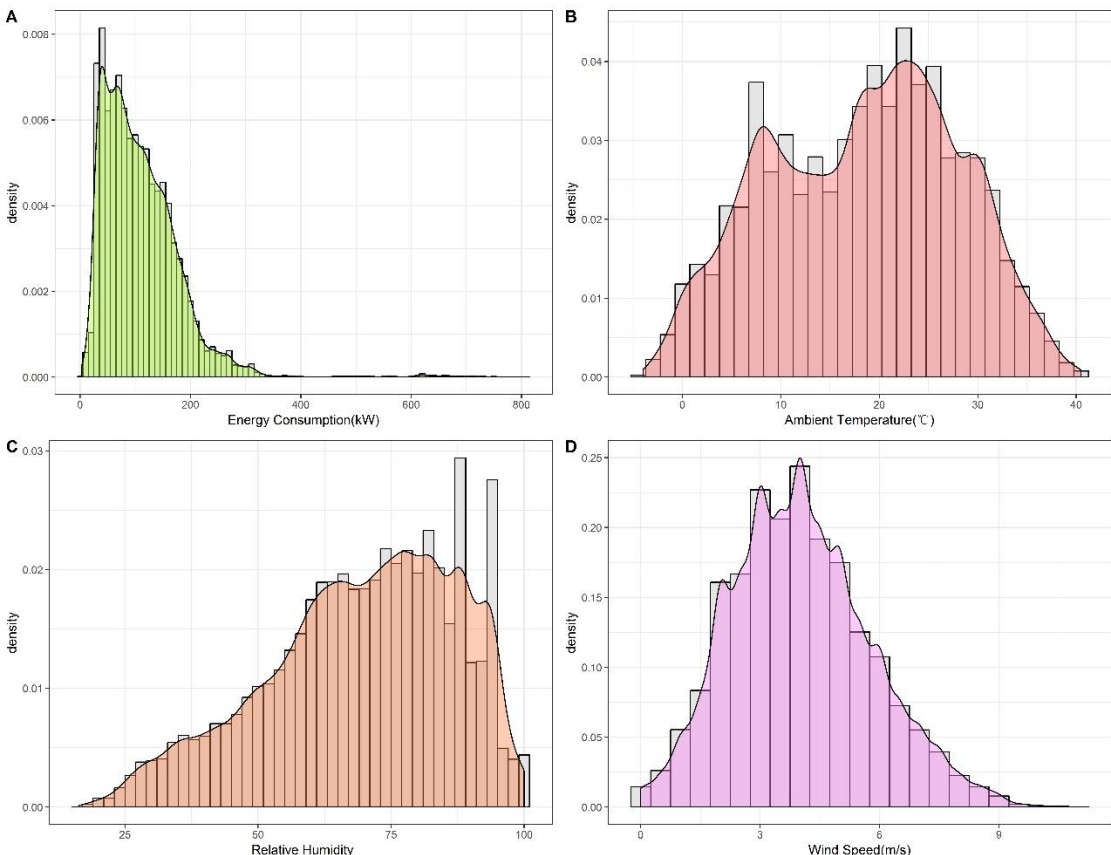

**Figure 1.** Probability distribution of the indoor load and weather data.

### 3.2. Research Outline

Figure 2 illustrates the outline of the research in this paper. It is divided into three main steps. The first step is data preparation, including energy consumption data and weather data, where the energy consumption data is calculated based on the temperature, humidity, and flow rate of the indoor air-conditioning inlet and outlet, while the weather data comes from airport data as described in Section 2.1.

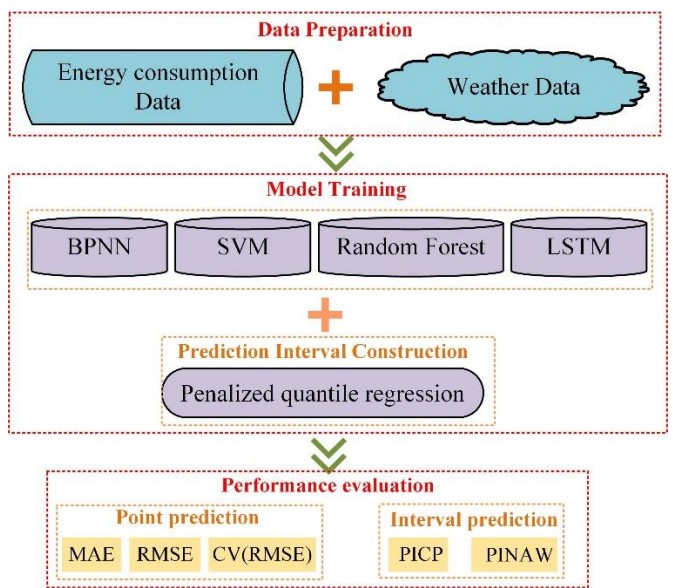

**Figure 2.** Research outline.

The second step is model training. In this paper, the deep-learning method, LSTM, is chosen as the basic point estimation algorithm, and the three commonly used machine-learning methods, BPNN, SVM, and random forest (RF), are used as reference models to facilitate comparisons with other models. Based on the prediction results of the above point estimation algorithms, penalized quantile regression (PQR) was chosen to construct the prediction intervals. Specifically speaking, as energy consumption data and weather data are input variables, we defined them as $x_t (t = 1,2, \dots, n)$. Thus, the outputs of the aforementioned machine-learning methods can be expressed as:

(1)　BPNN

We can describe the training process of BPNN by the following equations:

$$node_j = \sum_{t=1}^{n} \omega_{tj} x_t, j = 1,2, \dots, m \tag{1}$$

$$z_j = f_H(node_j), j = 1,2, \dots, m \tag{2}$$

Here, $node_j$ is the activation value of the jth node, $z_j$ is the output of the hidden layer, and $f_H$ is the activation function of a node, which is usually a sigmoid function:

$$f_H(x) = \frac{1}{1 + \exp{(-x)}} \tag{3}$$

The outputs of all neurons in the output layer are given as follows:

$$y_t = f_t(\sum_{j=1}^{m} \omega_{tj} z_j) \tag{4}$$

Here, $f_t$ is the activation function, usually a line function.

In this paper, the single hidden layer BPNN model is chosen for predictive modelling, considering its generalizability and relatively low computational cost.

(2)　SVM

SVM uses the following form to approximate the relationship between input and output variables:

$$f(x_i) = \omega \cdot \varphi(x_i) + b \tag{5}$$

where $\varphi(x_i)$ represents a high-dimensional feature space mapped from the low-dimensional space. In terms of the $\omega$ and b, the following regularized risk function is used to find them.

$$\frac{1}{2} \|\omega\|^2 + C \frac{1}{n} \sum_{i=1}^{n} L_\varepsilon(y_i, f(x_i)) \tag{6}$$

where the $\|\omega\|^2$ is called the regularized term. The $L_\varepsilon(y_i, f(x_i))$ is the empirical error measured by the $\varepsilon$-insensitive loss function, which is defined as follows:

$$L_\varepsilon(y_i, f(x_i)) = \begin{cases} 0, & y_i - f(x_i) \leq \varepsilon \\ |y_i - f(x_i)| - \varepsilon, & otherwise \end{cases} \tag{7}$$

This defines the range of ε values such that if the predicted value is within the range, the loss is zero, and if the predicted point is outside the range, then the loss is the magnitude of the difference between the predicted value and the distance ε of the range. $C$ is a penalty factor.

To get the estimation of ω and b, Equation (5) is transformed into objective function (8) by introducing the positive slack variables, $\xi, \xi^*$.

$$Minimize \frac{1}{2} \|\omega\|^2 + C \frac{1}{n} \sum_{i=1}^{n} L_\varepsilon(y_i, f(x_i))$$

$$\text{Subject to } \begin{aligned} & y_i - \omega \cdot \varphi(x_i) - b \leq \varepsilon + \xi, \\ & \omega \cdot \varphi(x_i) + b - y_i \leq \varepsilon + \xi^*, \\ & \xi, \xi^* \geq 0 \end{aligned} \tag{8}$$

By introducing four Lagrange multipliers,$\alpha_i$, $\alpha_i^*$, $\xi_i$, $\xi_i^*$, we can turn Equation (8) to the Lagrangian function. The Lagrange function becomes:

$$L = \frac{1}{2}\omega^2 + C\sum_{i=1}^{n}(\xi_i + \xi_i^*) - \sum_{i=1}^{n}(\eta_i\,\xi_i + \eta_i^*\xi_i^*) - \sum_{i=1}^{n}\alpha_i(\varepsilon + \xi_i - y_i + \omega \cdot \varphi(x_i) + b) - \sum_{i=1}^{n}\alpha_i^*(\varepsilon + \xi_i^* + y_i - \omega \cdot \varphi(x_i) - b) \tag{9}$$

By introducing kernel function, $K(x, x_i)$, the SVM forecasting model can be obtained by quadratic programming:

$$f(x) = \sum_{i=1}^{n}(\alpha_i - \alpha_i^*)K(x, x_i) + b \tag{10}$$

where $K(x, x_i)$ is the kernel function of the SVR model. The commonly used kernel functions of the SVR models contain Gaussian, polynomial, and sigmoid.

(3) RF

RF constructs a large number of decision trees during training and averages the values of output from each tree as the final output. Firstly, the training data are randomly selected and used to build the $B$ decision tree. Then, RF is generated by adding in parallel these $B$ trees $\{T_1(x), T_2(x), \dots T_B(x)\}$, where $x = \{x_1, x_2, \dots x_n,\}$ is an n-dimensional feature vector. The ensemble produces $B$ outputs $\{Y_1 = T_1(x), Y_2 = T_2(x), \dots Y_B = T_B(x)\}$, where $Y$ is the value predicted by the decision tree; here, the estimation process of each tree is totally independent. The final prediction,$y_t$, is made by averaging the predicted values of each tree.

$$y_t = \frac{Y_1 + Y_2 + \cdots + Y_B}{B} \tag{11}$$

(4) LSTM

The long short-term memory network is derived from recurrent neural networks (RNN), and it can handle the long-term dependency problems of RNN. Hochreiter and Schmidhuber [38] mathematically elaborated the architecture of LSTM, and the article showed that LSTM can keep or delete information by three controlling gates: input gate, output gate, and forget gate. Figure 2 shows a schematic diagram of a LSTM network; $y_{t-1}$ represents the output of the previous step, and $x_t$ is the input of the current step.

When an input, $x_t$, enters the LSTM, the forget gate,$f$, decides what information is deleted. This decision can be computed as:

$$f_t = \sigma(W_f \cdot [y_{t-1}, x_t] + b_f) \tag{12}$$

Here, $\sigma$ is the sigmoid activation function applied to elements inside the parentheses, $W_f$ is the weight matrix of the forget gate, and $b_f$ is the bias of the forget gate.

While for the input gate, $i$, it is used to decide what information to keep and update in the LSTM, and $i$ can be expressed as:

$$i_t = \sigma(W_i \cdot [y_{t-1}, x_t] + b_i) \tag{13}$$

Here, $W_i$ is the weight matrix of the input gate, and $b_i$ is the bias of the input gate.

Then, activation function, $\tilde{C}_t$, is used to describe the cell state under the current input:

$$\tilde{C}_t = tanh(W_C \cdot [y_{t-1}, x_t] + b_C) \tag{14}$$

After information selecting, the previous LSTM cell state, $C_{t-1}$, can be updated to the current LSTM cell state $C_t$ as shown in Figure 2, and $C_t$ can be computed as:

$$C_t = f_t * C_{t-1} + i_t * \tilde{C}_t \tag{15}$$

Here, $*$ is an element-wise multiplier.

The output gate can be used to scale the output of the LSTM activation function, and it is expressed as:

$$o_t = \sigma(W_o \cdot [y_{t-1}, x_t] + b_o) \tag{16}$$

Finally, the output, $y_t$, is calculated based on the LSTM cell state and output gate, $o$, and can be expressed as:

$$y_t = o_t * tanh(C_t) \tag{17}$$

(5)　Penalized quantile regression

Quantile regression is a modelling approach that models the quantile of the distribution of the variables. While traditional regression models focus on calculating the mean value of the target variable, quantile regression calculates the median value of it. Quantile regression utilizes a quantile loss function to provide information about future uncertainty:

$$L_{q,t}(y_t, p^q{}_t) = \sum_{i:y_t \le p^q{}_t}^N (1-q)(p^q{}_t - y_t) + \sum_{i:y_t > p^q{}_t}^N q(y_t - p^q{}_t) \tag{18}$$

where $q$ denotes the quantile, $y_t$ is the output at time $t$ obtained from the machine learning-methods, $p^q{}_t$ denotes the estimated $q_{th}$ quantile at time $t$, and $L_{q,t}$ denotes the quantile loss for the $q_{th}$ quantile at time $t$.

For the smoothness of the fitting result of quantile regression, we adopt an appropriate penalty function which is designed to penalize the difference of consecutive regression coefficients [39]. Suppose that the estimated $q_{th}$ quantile function is given as a linear function, $\sum_{j=1}^M \beta_j x_{ij}$, and the penalty function can be expressed as $\sum_{j=1}^M (\beta_{j+1} - \beta_j)^2$. This penalty function gives a smoothing effect of the regression coefficients by penalizing the high quantity of the difference of two adjacent coefficients. Then, the objective loss function is represented by:

$$\begin{aligned} L_{q,t}(y_t, \beta x) = \sum_{i:y_t \le \beta x}^N (1-q)(\beta x - y_t) + \sum_{i:y_t > \beta x}^N q(y_t - \beta x) + \\ \sum_{j=1}^M (\beta_{j+1} - \beta_j)^2 \end{aligned} \tag{19}$$

The third step is model performance evaluation, which is divided into two main aspects: firstly, the traditional point forecast evaluation criteria are used to assess the daily and weekly indoor-load-forecasting performance of each model, i.e., MAE, RMSE, and CV (RMSE). Secondly, to assess the interval forecasting results of the models, PICP and PINAW were used for quantitative comparison, and all interval forecasts were within the 95% prediction interval. The following subsection provides a brief description of the evaluation metrics used in this study.

### 3.3. Optimizing Hyperparameters for Machine-Learning Methods

In this section, a widely used optimisation algorithm, grid search, is introduced to find the optimal hyperparameters for machine-learning methods. In the hyperparametric optimisation process, the performance of the model needs to be tested by comparing the predicted energy consumption with actual energy data. However, a fixed partition of the training and testing sets may lead to severe overfitting problems. To eliminate possible overfitting, we use the k-fold cross-validation method (k = 5) for optimization. This technique obtains a comprehensive evaluation metric of the prediction model by transforming the training and testing sets multiple times, reflecting the predictive performance and generalization ability of the model.

In this study, the parameters in the BPNN, SVM, RF, and LSTM are selected to be optimized to improve the accuracy of the energy prediction models. Table 3 lists the range and optimal combination of hyperparameters for using the grid search with the cross validation.

**Table 3.** Optimal combination of hyperparameters using grid search with cross validation.

| Model | Hyperparameters | Range | Step Size | Optimal Value |
|---|---|---|---|---|
| BPNN | Number of neurons | [10,100] | 5 | 40 |
| | Activation function | 'sigmoid', 'tanh', 'relu' | None | 'sigmoid' |
| | Initial learning rate | [0.001,0.01] | 0.001 | 0.002 |
| SVM | Kernel function | 'Gaussian', 'linear', 'poly' | None | 'Gaussian' |
| | C | [1,5] | 0.5 | 3 |
| | Epsilon | [0.1,1] | 0.1 | 0.2 |
| RF | Number of the trees | [5,100] | 5 | 50 |
| | Depth of the trees | [10,120] | 10 | 50 |
| | Minimum sample number in leaf node | [1,5] | 1 | 1 |
| LSTM | Number of neurons | [10,100] | 5 | 40 |
| | Activation function | 'softsign', 'sgdm', 'tanh' | None | 'sgdm' |
| | Initial learning rate | [0.001,0.01] | 0.001 | 0.002 |

*3.4. Evaluation Metrics*

To evaluate the model accuracy of point estimation, we use three metrics: the root mean square error (*RMSE*), mean absolute error (*MAE*), and coefficient of variation of *RMSE* (*CV(RMSE)*). All these error evaluation indexes have been extensively applied in the forecasting model estimation. *RMSE* and *MAE* can be expressed as below:

$$RMSE = \sqrt{\frac{1}{n}\sum_{i=1}^{n}(p_i - o_i)^2} \tag{20}$$

$$MAE = \frac{1}{n}\sum_{i=1}^{n}|p_i - o_i| \tag{21}$$

where, $p_i$ is the predicted value, $o_i$ is the observed value, and $n$ is the number of measured data.

*CV(RMSE)* is the measure of the accumulated magnitude of error and is calculated as follows:

$$CV(RMSE) = \frac{\sqrt{\frac{1}{n}\sum_{i=1}^{n}(p_i - o_i)^2}}{\frac{1}{n}\sum_{i=1}^{n}o_i} \tag{22}$$

For assessing the interval prediction results, two evaluation metrics obtained from [40] can be utilized in this work. Firstly, the prediction interval coverage probability (*PICP*) evaluates whether the actual value of energy consumption is within the predicted interval. In addition, the prediction interval normalized average width (*PINAW*) is used to measure the width of the predicted interval. The *PINAW* is related to the informativeness of *PICP* or equivalently to the sharpness of the predictions. If *PINAW* is large, the *PICP* will have little value as it is meaningless to say that future energy consumption will lie within its possible extreme range. Ideally, prediction intervals should have *PICPs* close to the expected coverage rate and low *PINAWs*. These two metrics are defined as below:

$$PICP = \frac{1}{n}\sum_{i=1}^{n}\alpha_i \tag{23}$$

$$PINAW = \frac{1}{nE}\sum_{i=1}^{n}(U_i - L_i), \tag{24}$$

where $\alpha_i$ = 1 if the actual energy consumption lies within the prediction interval, and $\alpha_i$ = 0 otherwise. $L_i$ and $U_i$ represent the minimum and the maximum values of the predicted interval, respectively. $E$ is the difference between them.

To comprehensively compare the different evaluation metrics, the entropy weights method (*EWM*), which is a commonly used information-weighting method in decision making [41], is introduced in this paper. The *EWM* is mainly divided into the following steps:

Step 1: Construction of the initial matrix

The initial matrix is constructed as follows:

$$X = \begin{bmatrix} x_{11} & x_{12} & \cdots & x_{1m} \\ x_{21} & x_{22} & \ddots & x_{2m} \\ \vdots & & \ddots & \vdots \\ x_{n1} & x_{n2} & \cdots & x_{nm} \end{bmatrix} \tag{25}$$

where $X$ is the initial matrix, $m$ is the number of factors, $n$ is the number of data sample, $x_{ij}$ is the analyzed values of each sample parameter, $i$ = 0, 1, 2, ..., $m$ and $j$ = 0, 1, 2, ..., $n$.

Step2: Normalization of the matrix

The normalized matrix can be expressed as following:

$$X' = \begin{bmatrix} x'_{11} & x'_{12} & \cdots & x'_{1m} \\ x'_{21} & x'_{22} & \ddots & x'_{2m} \\ \vdots & & \ddots & \vdots \\ x'_{n1} & x'_{n2} & \cdots & x'_{nm} \end{bmatrix} \tag{26}$$

$$x'_{ij} = \frac{(x_{ij})^j_{max} - x_{ij}}{(x_{ij})^j_{max} - (x_{ij})^j_{min}}$$

where $X'$ is the normalized decision matrix.

Step3: Calculation of the entropy

The entropy of each factor is calculated as the following:

$$e_j = -\frac{1}{\ln n}\sum_{i=1}^n p_{ij}\ln p_{ij}$$

$$p_{ij} = x'_{ij}/\sum_{i=1}^n x'_{ij} \tag{27}$$

where $e_j$ is the entropy of each factor.

Step 4: Calculation of the weight

The weight of each factor is calculated as the following:

$$w_j = (1 - e_j)/\sum_{j=1}^m (1 - e_j) \tag{28}$$

where $w_j$ is the weight of each factor.

Step 5: Calculation of the composite score

The composite score of factors can be calculated as the following:

$$s_i = \sum_{j=1}^m w_j x_{ij} \tag{29}$$

where $s_i$ is the composite score of the factors.

## 4. Results and Discussion

### 4.1. Model Training

The first step to setup a prediction model is to determine the number of lag periods, which can also be understood as input to the time-series model. AIC and BIC have been widely used in literature [42,43]; these two criteria can find the best balance between the complexity of the model and the accuracy of the model. Therefore, we select AIC and BIC to be the selection criteria for the number of lag periods, and they are implemented using MATLAB. Figure 3 shows the variations of AIC and BIC in the different number of lag periods.

As seen in Figure 3, both criteria present a gradual decline in the trend, while the value of AIC rises slowly after each sharp drop when the number of lag periods is greater than 100. The software EViews is also applied to ensure the reliability of the results as it is very convenient to calculate AIC and BIC. After comprehensive consideration of two results, 292 is chose to be the optimal number of lag periods, which is indicated by a dotted purple line in Figure 3.

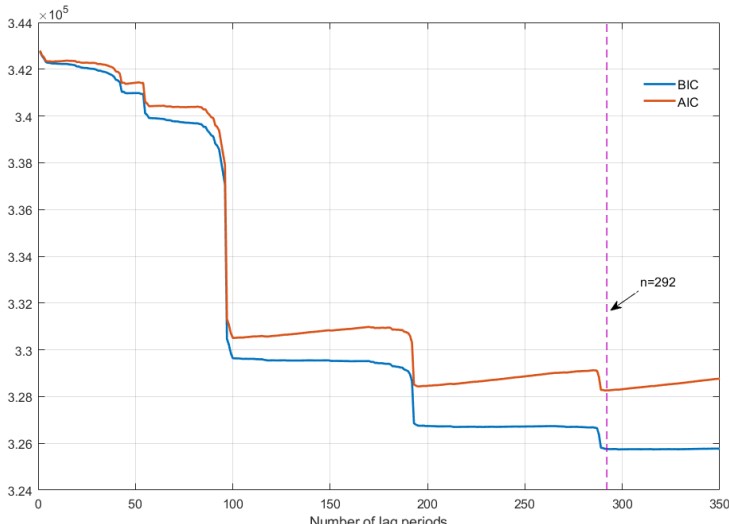

**Figure 3.** Variations of AIC and BIC.

After obtaining the optimal lag periods, we adopt an iterative approach for predicting the daily and weekly ahead indoor loads. The general idea is illustrated in Figure 4. Firstly, the prediction models are performed based on the historical values, and the predicted value at the next time step is generated by the prediction models. Then, this predicted value is combined with part of the historical values to form the new inputs. Finally, the process stops till all predictions of the daily or weekly indoor loads are generated. Therefore, the same lagged values are used for one week and one day ahead forecasting.

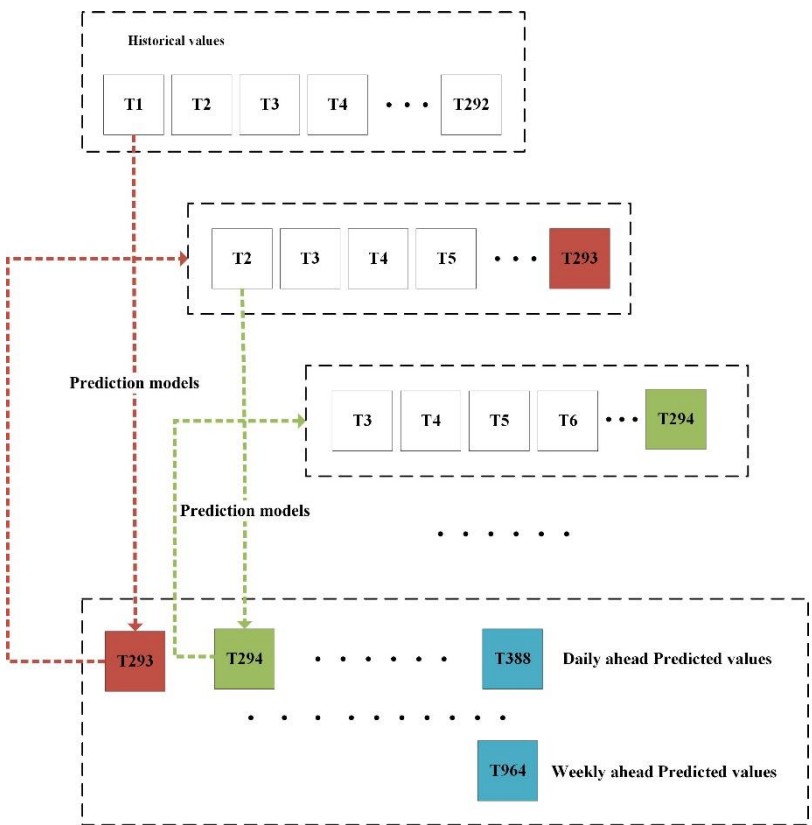

**Figure 4.** Iterative process for daily and weekly ahead indoor load prediction.

*4.2. Point Prediction Results*

In this section, the results of the point prediction performances of the models are evaluated on the test data. It is worth noting that the test data contains continuous and complete data for 112 days; therefore, we obtain 112 days of results for one day ahead forecasting, and each day contains 96 predicted values and 16 weeks of results for one week ahead forecasting, and each week contains 672 predicted values, respectively. It can be noticed that there are measurable differences between the results of the MAE of the test models for daily and weekly load forecasting in Figure 5. In general, RF performs the worst for one day ahead forecasting and SVM performs the worst for one week ahead forecasting in terms of MAE, and the performance difference between LSTM and BPNN is not obvious. Nevertheless, in Figure 5A,B, the median MAE of LSTM is still lower than that of BPNN. It can be indicated that the LSTM model is more advantageous and reliable than the other models.

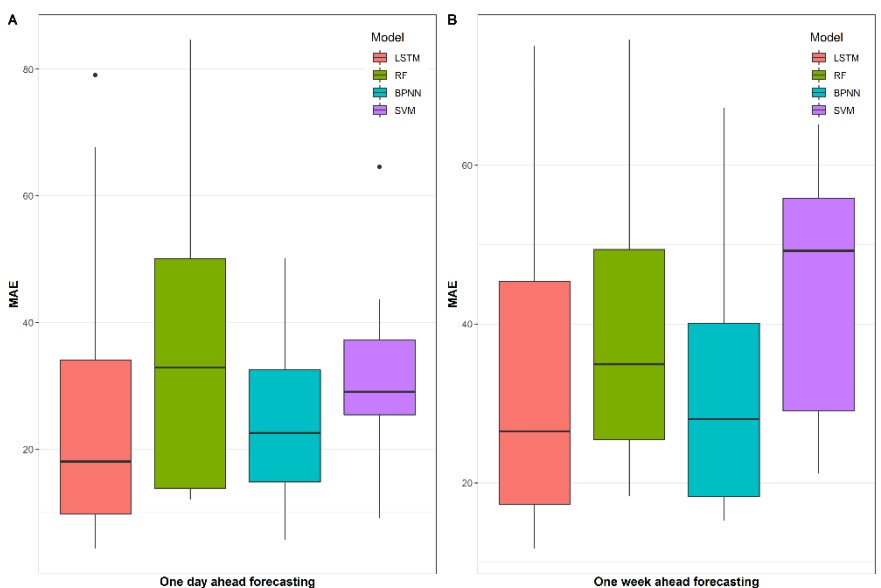

**Figure 5.** MAE results of proposed models for daily and weekly load forecasting.

Figures 5 and 6 depict the RMSE results of the proposed models for daily and weekly load forecasting. Obviously, the performances of the other three models are still not as good as the LSTM model. It is interesting to note that there are some extremely high values in the RF and LSTM models, especially in Figure 6A. A possible reason for this might be due to the failure of the model in some situation as the predicted error increases with the number of iterations. We also observe that the difference between the RMSE of LSTM and other models are amplified in Figure 6, which further indicate that the LSTM model outperforms other models due to the smaller dispersion degree of RMSE distribution.

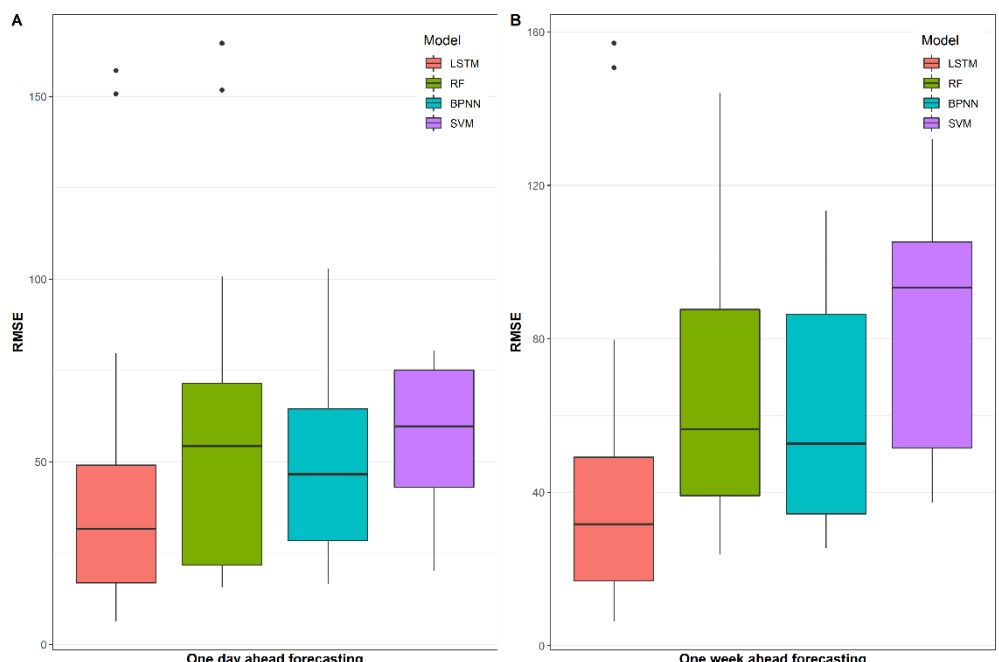

**Figure 6.** RMSE results of proposed models for daily and weekly load forecasting.

Regarding the CV(RMSE), as illustrated in Figure 7, the differences between the proposed models are more evident. SVM performs badly in one day ahead load forecasting, as the CV(RMSE) of this model is much higher than that of other models. In addition, an interesting finding can further be noticed. In general, the overall distribution

of CV(RMSE) in one day ahead load forecasting is higher than that in one week ahead load forecasting, which indicates that the prediction of the energy consumption trend in the short-term time is relatively accurate.

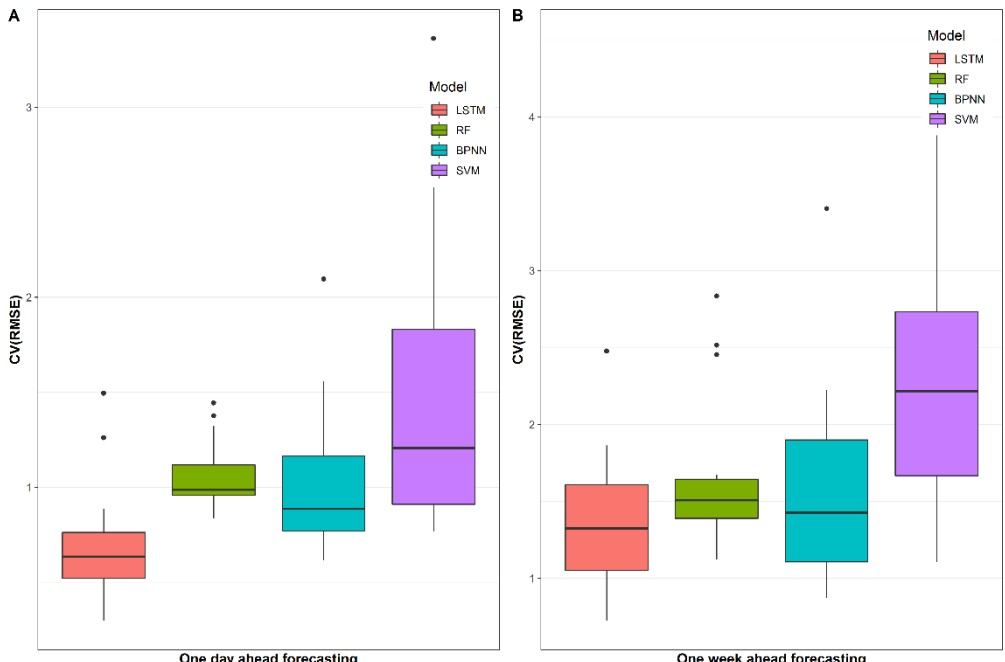

**Figure 7.** CV(RMSE) results of proposed models for daily and weekly load forecasting.

Based on the point prediction results above, it can be concluded that the LSTM models perform the best among these commonly used prediction models. The reason for this may be that LSTM learned the dependencies between input variables in sequence by its chain structure, i.e., the periodic characteristics of the energy consumption data in the time domain. Therefore, in terms of periodic prediction time, LSTM will have an advantage over other commonly used models. It is worth noting that although the LSTM model obtains the best prediction performance among the three proposed models, it still has a large gap in its performance compared with short-time load forecasting [44]; it is inevitable because the prediction errors add up as the number of prediction steps increases. Table 4 indicates that the LSTM models perform the best among these commonly used prediction models as well.

**Table 4.** Composite score of proposed models for daily and weekly load forecasting.

| Model | Composite Score for Daily Load Forecasting | Composite Score for Daily Load Forecasting |
|---|---|---|
| LSTM | **99.60** | **99.60** |
| RF | 16.52 | 60.59 |
| BPNN | 57.32 | 81.14 |
| SVM | 8.94 | 0.20 |

*4.3. Interval Prediction Results*

In this section the results of the interval prediction are obtained by the aforementioned four models integrated with PQR. The forecast results of 16 weeks of the test data are presented in Figures 8 and 9B. By comparing the four proposed methods, it can be seen that the width of the prediction interval of RF is the largest, whereas the probability that the actual value falling into the calculated prediction interval limits of SVM is the lowest. On the contrary, LSTM obtains the highest PICP and lowest PINAW

among the proposed models, which means this model is suitable for weekly interval load forecasting. Furthermore, the numerical results for the weekly interval load forecasting are described in Table 5; it is found that the prediction interval of LSTM and RF can cover nearly all the actual load in some weeks' predictions, as the maximum PICP of these models is very close to 1. In terms of PICP in Table 5, after comprehensive comparison, the LSTM model still has the best performance, although the standard deviation of the PICP results is not the smallest. Moreover, in the aspect of the width of the prediction interval, there is a slight advantage in viewing the PINAW results in the LSTM model compared with the BPNN model. However, our main concern is still the accuracy of the prediction, and thus, PICP accounts for a larger share of the forecast results. Therefore, LSTM-PQR is proven to be a more convincing model for weekly interval load forecasting.

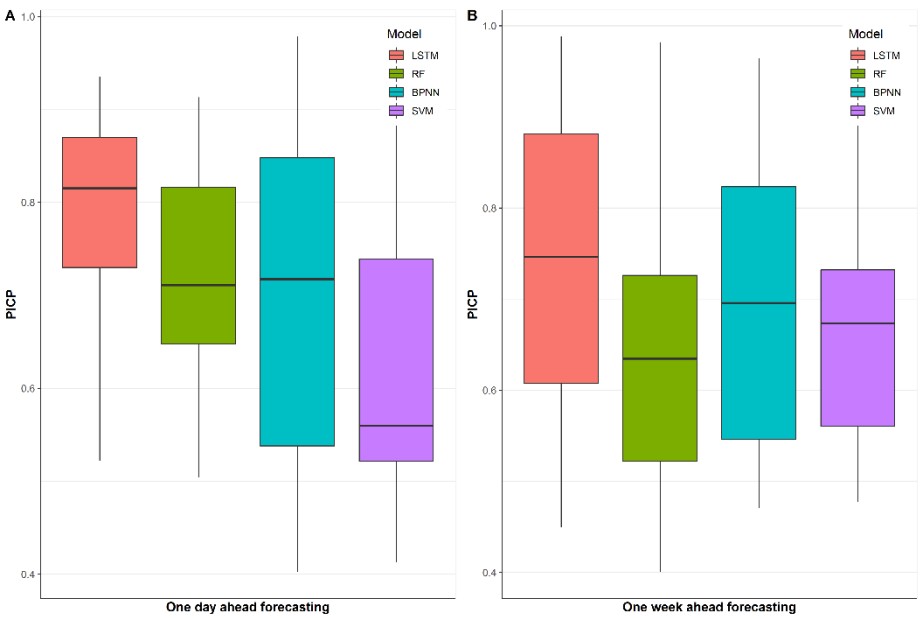

**Figure 8.** PICP results of proposed models for daily and weekly interval load forecasting with 95% prediction interval.

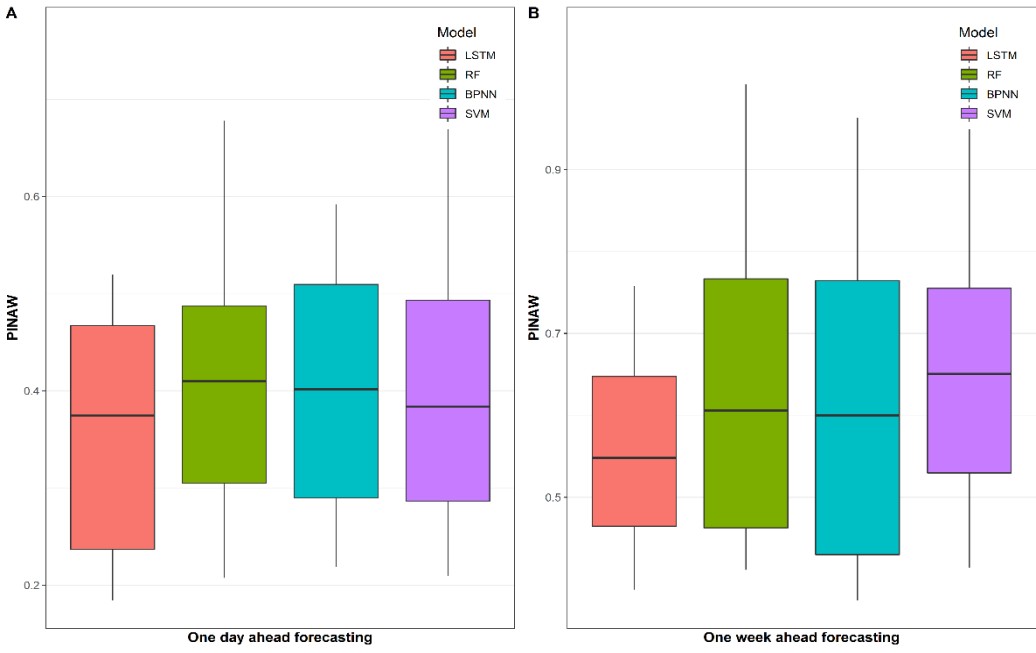

**Figure 9.** PINAW results of proposed models for daily and weekly interval load forecasting with 95% prediction interval.

**Table 5.** One week ahead prediction accuracy.

| Model | PICP (mean) | PICP (max) | PICP (min) | PICP (SD) | PINAW (mean) | PINAW (max) | PINAW (min) | PINAW (SD) |
|---|---|---|---|---|---|---|---|---|
| LSTM | **0.739** | **0.988** | 0.450 | 0.171 | **0.565** | **0.757** | **0.386** | **0.123** |
| RF | 0.646 | 0.982 | 0.400 | 0.179 | 0.637 | 1.004 | 0.411 | 0.194 |
| BPNN | 0.687 | 0.964 | 0.470 | 0.158 | 0.618 | 0.963 | 0.399 | 0.192 |
| SVM | 0.669 | 0.949 | **0.523** | **0.127** | 0.659 | 1.063 | 0.413 | 0.186 |

Similar to the weekly interval load forecasting, the results of the daily interval load forecasting are shown in Figures 8 and 9A. In the mass, the prediction performance of the proposed models for daily load is improved in terms of PICP; it is reasonable due to the relative simplicity of the short-term model. Likewise, LSTM-PQR maintains the best predictive performance, while SVM has the lowest value of PICP, and RF gets the highest value of PINAW.

As exhibited in Table 6, LSTM performs better than the other three proposed models, and the mean PICP of LSTM is 6.4% and 20.9% higher than that of RF and SVM, respectively. Despite that, BPNN has the highest value of maximum PICP among these models; it is still not an optimal model as it has the highest value of standard deviation PICP. In addition, the standard deviation PICP of the LSTM model is similar to the RF model but inferior to RF model. This is due to the stable point prediction performance of the RF model, which can be seen in Figure 7A. It can be concluded that the stability of the point prediction affects the stability of the interval prediction to some extent.

**Table 6.** One day ahead prediction accuracy.

| Model | PICP (mean) | PICP (max) | PICP (min) | PICP (SD) | PINAW (mean) | PINAW (max) | PINAW (min) | PICP (SD) |
|---|---|---|---|---|---|---|---|---|
| LSTM | **0.781** | 0.935 | **0.522** | 0.113 | **0.356** | **0.519** | **0.184** | **0.121** |
| RF | 0.731 | 0.913 | 0.504 | **0.111** | 0.414 | 0.608 | 0.207 | 0.142 |
| BPNN | 0.694 | **0.978** | 0.402 | 0.197 | 0.403 | 0.592 | 0.218 | 0.126 |
| SVM | 0.618 | 0.911 | 0.413 | 0.151 | 0.410 | 0.766 | 0.209 | 0.158 |

The prediction performance advantage also further illustrates that the LSTM-PQR model can accurately predict the indoor load of the building range within a period of time. Similar to the weekly interval load forecasting, LSTM-PQR get the highest value of PICP and the lowest value of PINAW simultaneously; this indicates that LSTM-PQR can lead to accurate and reliable results in the daily interval load forecasting. The results in Table 7 also confirm this conclusion.

**Table 7.** Composite score of proposed models for daily and weekly interval load forecasting.

| Model | Composite Score for Weekly Interval Load Forecasting | Composite Score for Daily Interval Load Forecasting |
|---|---|---|
| LSTM | **99.60** | **99.60** |
| RF | 11.95 | 21.27 |
| BPNN | 43.78 | 27.45 |
| SVM | 12.37 | 4.96 |

## 5. Conclusions

In this paper, we applied the deep-learning method, LSTM, with penalized quantile regression for interval indoor load forecasting in an office building. The commonly used machine-learning methods, BPNN, SVM, and random forest, are adopted as reference models. The point prediction performance and interval prediction performance of the proposed models have been comprehensively studied in the paper. We can draw the following conclusions:

1.  The proposed LSTM-PQR model has better performance than BPNN, SVM, and RF for interval indoor load forecasting in an office building.
2.  For point prediction performance, the distribution of MAE and RMSE in different models are quite different. Generally, the LSTM model maintains optimal performance, while the SVM and RF models have relatively poor prediction results.
3.  For interval prediction performance, the LSTM-PQR model is still the most suitable choice, especially in daily interval load forecasting, and it has improvements ranging from 6.4% to 20.9% for PICP comparing with the other three models.

Future works should focus more on the ability to capture sudden sharp increments of the load, as it is quite challenging work and there is still no literature on this area, and the prediction of negative energy consumption, such as in eco-parks where buildings receive and deliver energy to factories [45].

**Funding:** This research received no external funding.

**Data Availability Statement:** The data presented in this study are available on request from the corresponding author. The data are not publicly available due to commercial privacy.

**Conflicts of Interest:** The authors declare no conflict of interest.

## Nomenclature

| | |
|---|---|
| ANN | Artificial neural network |
| SVM | Support vector machine |
| DT | Decision tree |
| PSO | Particle swarm optimization |
| PCA | Principal component analysis |
| MAE | mean absolute error |
| RMSE | root mean square error |
| AIC | Akaike Information Criterion |
| BIC | Bayesian Information Criterion |
| CV(RMSE) | Coefficient of variation of the RMSE |
| PQR | Penalized quantile regression |
| RF | Random forest |
| PICP | Prediction interval coverage probability |
| PINAW | Prediction interval normalized average width |
| DE | Differential evolution |

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
