# Peer review of "A Novel Interval Energy-Forecasting Method for Sustainable Building Management Based on Deep Learning"

_sustainability, doi:10.3390/su14148584_

Round 1

Reviewer 1 Report

Some comments as follows:

(1) For the input variables x_t in Line 166, specify it in Line 327 when the optimal lag is chosen as 292. There are four attributes used in this study as indicated in Figure 1. So the input variables x_t would be 4*292? Correct it if it is wrong.

(2) What about the calendar information that may affect energy consumption? The calendar information can be easily obtained and often used together with the weather data, e.g., Deep Learning Based Short‐Term Load Forecasting Incorporating Calendar and Weather Information. Why not add the calendar information, e.g., weekday or weekends/holidays?

(3) Are you using the same lag value for one week and one day ahead forecasting? If yes, why not check AIC and BIC separately for these two cases?

(4) As indicated in Line 136, the time interval in the data is 15 minutes. While in Line 345, you obtain 112 results for 112 days. This is inconsistent since you may get 24 hours / 15 minutes = 96 results for a single day. Clarify this and fix the inconsistency.

Reviewer 2 Report

You have a good job done, please amend some points presented in the attachment and I believe you can have a good pice in your hands. BR

Round 2

Reviewer 1 Report

Dear authors,

The previous concerns are well answered, but the revisions are not reflected in the manuscript. For example, it is still the saying that "we obtain 112 results for one day ahead forecasting and 345 16 results for one week ahead forecasting respectively". The authors should add the discussion for previous comments 2-4 into the manuscript. 

Reviewer 2 Report

Ok, done

Author Response

Dear reviewer:

Greetings and good day!

We would like to express our sincere gratitude to your effort in reviewing our manuscript and giving valuable suggestions/criticisms and proposing important revisions. Thank you for your affirmation of the revised manuscript.

Yours sincerely,

Yun Duan

This manuscript is a resubmission of an earlier submission. The following is a list of the peer review reports and author responses from that submission.

Round 1

Reviewer 1 Report

Main comments:

  • Do all models use the same input variables ? Has some research been done to select them, such as sensitivity analysis ? Are other weather variables (wind speed, solar irradiance) considered irrelevant ?
  • Is it relevant to have the flow rate of indoor air conditioning described as an input variable, since it can be controlled ?
  • Energy consumption often has a temperature-independent periodic component. How is this handled by LSTM ?

Minor comments :

  • acronyms should be explicited at their first use. I suggest moving the nomenclature from the appendix to the front of ther paper.
  • some equations (1, 2, 3, 5) have a formatting issue
  • the numbering of sections should be reviewed (conclusion is numbered 1)

Reviewer 2 Report

This paper presents a good research effort. Some comments  can improve the paper quality as: 

  1. Don't use an abbreviation without a full definition for the first usage see for example BPNN, SVM, LSTM...etc.
  2. Rearrange the introduction section as motivation, literature review, research gap, contribution then the paper organization.
  3.  Discuss the modeling of data uncertainty in the target problem 
  4. Simulation results need more discussion 
  5. In conclusion. concentrate on your actual findings and the extension of your studies. 
  6. Typo errors must be avoided.  

Reviewer 3 Report

Dear authors,

This manuscript is not ready for publication for the following reasons:

(1) The major concern is about the novelty and contribution of this study, since LSTM and PQR are existing methods.

(2) The main contributions as well as the main findings should be summarized and added in the end of Introduction section.

(3) Section 2.3 and 2.4 are both general introductions, without connecting to the specific problem the author wants to solve in this study.

(4) The hyperparameters and other detailed experiment settings are not given in the Results and Discussion section.

(5) A separate Related Work section should be added, from the content currently discussed in the Introduction section.

(6) Some abbreviations are used in the abstract directly without giving the formal definition first, e.g., BPNN, SVM, LSTM, PQR and PICP.

(7) The math symbol does not display correctly in Equations (1)-(3), (5).

(8) The section title "1. Conclusion" should be re-numbered as "4. Conclusion"

Round 2

Reviewer 1 Report

The reviewers' comments seem to have been taken into account. The paper shows a principled study and may be published as such, although there is little scientific originality.

Author Response

Dear reviewer:

Greetings and good day!

Thank you for your valuable comments on our manuscripts and we will continue to improve our scientific innovation in our future research work.

Yours sincerely,

Yun Duan

Reviewer 2 Report

No other comments.

Author Response

Dear reviewer:

Greetings and good day!

We would like to express our sincere gratitude to your effort in reviewing our manuscript and giving valuable suggestions/criticisms and proposing important revisions.

Yours sincerely,

Yun Duan

Reviewer 3 Report

1. The details are not enough in Section 2.2 with a vague flowchart in Figure 2. The similar way of mathematical formulations as Sections 2.3 and 2.4 is required for your method in Section 2.2.

2. Check the pdf version for Equations (1)-(4) where the symbols overlap.

3. Some abbreviations are still not explained in the main text (instead of the table) for the first time when they appear, e.g., BPNN.

Round 3

Reviewer 3 Report

Dear authors,

(1) I would suggest that you remove the general sections 2.3 and 2.4 and re-orgaize all the mathematical formulas in section 2.2, with the symbols defined for your specific energy forecasting problem. For example, x_t for input energy data and a new symbol for weather data, so that the readers can know what are the inputs for LSTM, BPNN, SVM, Random Forest in your work. Then their outputs are defined as y_t and used as input for quantile regression. Finally, the output of quantile regression is defined as p_i.

(2) What are the hyper-parameters for BPNN, SVM, and Random Forest?
